# Identification of jet lubrication oil as a major component of aircraft exhaust nanoparticles

Akihiro Fushimi[1], Katsumi Saitoh[1,2], Yuji Fujitani[1], Nobuyuki Takegawa[3]

[1]National Institute for Environmental Studies, Tsukuba 305-8506, Japan
[2]Environmental Science Analysis and Research Laboratory, Iwate 028-7302, Japan
[3]Tokyo Metropolitan University, Tokyo 192-0397, Japan

*Correspondence to*: Akihiro Fushimi (fushimi.akihiro@nies.go.jp)

**Abstract.** Jet engine aircraft are ubiquitous and significant sources of atmospheric nanoparticles. Using size-resolved
particulate samples collected near a runway of the Narita International Airport, Japan, we clearly demonstrate that organic compounds in the ambient nanoparticles (diameter: <30 nm) were dominated by nearly intact forms of jet engine lubrication oil. This finding provides direct evidence for the importance of unburned lubrication oil as a source of aircraft exhaust nanoparticles and also has an implication for their environmental impacts near airports and in the upper troposphere.

## 1 Introduction

Jet engine aircraft are a significant source of atmospheric nanoparticles and exist ubiquitously from ground level to the upper troposphere (Masiol and Harrison, 2014). A new particulate emission standard for turbofan and turbojet aircraft engines will come into effect from 2020 (International Civil Aviation Organization, 2017). Therefore, research needs to characterize aircraft exhaust particles have been increasing. Previous studies have shown that the impacts of aircraft on the distribution of nanoparticles in ambient air may range over a horizontal scale of approximately 16 km near airports (Hudda et al., 2014).
Nanoparticles can penetrate deep into the human respiratory tract and may have adverse effects on human health (Oberdorster et al., 2000; Biswas and Wu, 2005). The nanoparticles in diesel vehicle exhaust comprise mainly of organic compounds with relatively high saturation vapor pressures (alkanes, alkenes, etc.), and their lifetimes may be shorter than those of submicron particles owing to their evaporation and coagulation (Fushimi et al., 2008; Harrison et al., 2016). To understand the mechanisms of formation of nanoparticles emitted from jet engines and their physical/chemical
transformation in ambient air, it is important to determine the size distribution and chemical composition of nanoparticles emitted from in-use commercial aircraft under real-world conditions.

Previous studies have suggested that aircraft exhaust nanoparticles mainly comprise volatile particles. For example, the fraction of particles that completely evaporate below 300 °C are approximately 70–95%, and the number concentrations below 20 nm was found to decrease with an increasing ambient temperature (Wey et al., 2006). Bulk-level chemical analyses
of aircraft exhaust particles showed that the particle compositions were dominated by organic compounds under low engine thrust (e.g., idle and taxi) conditions and elemental carbon under high engine thrust (e.g., take-off and climb) conditions (Agrawal et al., 2008; Presto et al., 2011; Masiol and Harrison, 2014; Yu et al., 2017). This feature is contradictory to the volatility of aircraft exhaust nanoparticles. Sulfur compounds originating from jet fuels are also known to be the major components of aircraft exhaust particles (Masiol and Harrison, 2014; Yu et al., 2017). Particle emission factors show a
strong dependence on the contents of sulfur and aromatics in jet fuel (Wey et al., 2006; Masiol and Harrison, 2014). On the other hand, some qualitative markers of jet engine lubrication oil were commonly found in aircraft exhaust particles (Timko et al., 2010; Yu et al., 2012). The contribution of lubrication oil to the total organic mass may range from 5% to 100% (Timko et al., 2010; Yu et al., 2012). While such previous studies have provided useful insights into the characteristics of

aircraft exhaust particles, little is known regarding the origin and detailed chemical composition of size-resolved particles and especially nanoparticles (Kinsey, 2009; Presto et al., 2011; Kinsey et al., 2011; Masiol and Harrison, 2014).

The purpose of this study is to determine the size-resolved chemical composition of particles emitted from jet engine aircraft during take-off and landing in real-world conditions, and to estimate the contribution of jet fuel and lubrication oil to the nanoparticle mass. We have therefore collected particulate samples from an area near a runway at the Narita International Airport, Japan, during the daytime and nighttime using low-pressure cascade impactors. We used thermal-desorption gas chromatography mass spectrometry (TD-GC/MS) to identify the chemical composition of nanoparticles having diameters smaller than approximately 30 nm originating from real-world aircraft emissions, which is unprecedented.

## 2 Methods

### 2.1 Jet fuels and lubrication oils

To investigate the emission sources of the nanoparticles in aircraft exhausts, we obtained two Jet A1 fuels and two jet engine lubrication oils (Mobil Jet Oil II and Mobil Jet Oil 254, ExxonMobil, Irving, TX, USA) from Ishinokoyu Co. Ltd. (Mitaka, Tokyo, Japan). The Mobil Jet Oil II has a market share of 49% (Winder and Balouet, 2002), and the Mobil Jet Oil 254 is a newer generation oil.

### 2.2 Measurement site

Field measurements were conducted at a distance of 140 m west of runway A of the Narita (Tokyo) International Airport, Japan in winter (February 5–26, 2018, Fig. 1). The instruments used for obtaining the particle number size distributions and size-resolved particulate samplers were installed in a container. The airport has two runways. The runways A and B had an average of 401 and 290 flights per day, respectively, in 2017. Aircraft operations are allowed from 6:00 to 23:00. A summary of the aircraft models used in runway A of Narita International Airport during our measurement period is given in Table S1.

### 2.3 Particle number size distribution

The size distributions of the particle number concentration were measured every 1 s using the engine exhaust particle sizer (EEPS, Model 3090, D = 0.006–0.560 µm, TSI, Shoreview, MN, USA; flow rate: 10 L min$^{-1}$) during the entire measurement period. A copper tube (inside diameter: 10 mm, and length: 2 m), electrically conductive tubes (inside diameter: 6.35 and 9.53 mm, total length: 1 m; Part 3001788, TSI), and a glass manifold (inside diameter: 40 mm, total length: 600 mm) were used to transport the ambient air at a ground height of 3 m to the EEPS. To avoid particle deposition onto the sampling tubes and to increase the response speed, an extra pump (flow rate: 30 L min$^{-1}$) was used to vacuum the air inside the glass manifold.

The size distributions of the particle number were also measured every 5 min using a scanning mobility particle sizer (SMPS, Model 3936, D = 15–660 nm, TSI, Shoreview, MN, USA), which consists of an electrostatic classifier (Model 3080), differential mobility analyzer (DMA, Model 3081), and a condensation particle counter (CPC, Model 3022A). Ambient air was aspirated from the roof of the container through a stainless-steel tube (inner diameter: 10 mm, length: approximately 3 m, inlet height: approximately 3 m above ground level) and was split into the main sample flow (approximately 0.8 L min$^{-1}$) and bypass flow (approximately 20 L min$^{-1}$). The bypass flow was used to reduce the possible loss of nanoparticles due to Brownian diffusion. The main sample flow was diluted with particle-free air (approximately 2 L min$^{-1}$) and further split into the individual sample flows for the SMPS (approximately 0.3 L min$^{-1}$) and other CPCs (approximately 2.5 L min$^{-1}$). We used the dilution flow to reduce the effects of particle coincidence in the CPCs at higher concentrations. For the SMPS, the

penetration efficiency through the sampling tube was estimated to be >70% above 10 nm based on the theoretical formulae of Gormley and Kennedy (1949).

## 2.4 Sampling

Size-resolved particles in the ambient air at a height of approximately 3 m from the ground were collected using two low-pressure cascade impactors (NanoMoudi II, Model 125B, MSP, Shoreview, MN, USA; flow rate: 10.2 L min$^{-1}$) simultaneously. To distinguish the effect of aircraft emissions, the samples were collected during the daytime (during aircrafts operation hours) and nighttime (during non-operation hours). Three daytime (7:00–22:00) samples were collected from 16:24 on February 9, 2018 to 13:09 on February 13, 2018 (duration = 56.8 h, sample #1), from 17:33 on February 13, 2018 to 9:37 on February 17, 2018—except for 13:30–16:45 on February 15, 2018 (duration = 48.8 h, sample #2)—and from February 19 to February 20, 2018 (duration = 30.0 h, sample #3). One nighttime (0:00–6:00) sample was collected during February 22–26, 2018 (duration = 30.0 h, sample #4).

For one of the two NanoMoudi II impactors, a gold (Au) foil (diameter: 47 mm, Mitsubishi Materials, Tokyo, Japan; "NanoMoudi II-Au", hereafter) was used as the collection substrate for the impaction stage, and a quartz-fiber filter (diameter: 47 mm, 2500QAT-UP, Pall, East Hills, NY, USA) was used as the substrate for the backup filter. In the other NanoMoudi II, a polycarbonate membrane filter (Nuclepore; pore size: 0.05 μm; diameter: 47 mm; Whatman, GE Healthcare UK Ltd., Buckinghamshire, UK; "NanoMoudi II-PC," hereafter) was used as the collection substrate for the impaction stage. At each impaction stage, a nitrocellulose membrane filter (AAWP04700, pore size: 0.8 μm; diameter: 47 mm; Merck Millipore, Billerica, MA, USA) was set underneath the PC filter. A polytetrafluoroethylene with non-woven fabric polyethylene/polyethylene terephthalate membrane filter (TFH-47; diameter: 47 mm, Horiba, Kyoto, Japan, for samples #1 and #2) or a quartz-fiber filter (diameter: 47 mm, 2500QAT-UP, Pall, for samples #3 and #4) was used as the substrate for the backup filter. The use of different substrates as backup filters does not alter the flow rates. Using NanoMoudi II, the particles were separated into 14 size fractions. For the NanoMoudi II-Au, for example, the equivalent cut off aerodynamic diameters at a 50% efficiency (D50) of the impaction stages, calibrated and reported by the manufacturer, were as follows: 0.010, 0.018, 0. 032, 0.057, 0.105, 0.170, 0.290, 0.560, 1.00, 1.80, 3.10, 6.20, and 9.90 μm. Au foils were rinsed with acetone (dioxin analytical grade, Wako Pure Chemical Industries, Osaka, Japan) before use. A copper tube (inside diameter: 10 mm, length: 3 m) was used as the sampling line. Before each sampling, the impactor nozzles and the support rings of the NanoMoudi II were cleaned using acetone and blown off with an air-duster.

The NanoMoudi II-Au samples were used for particle-mass weighing, elemental/organic carbon (EC/OC) analysis, and organic analysis. The NanoMoudi II-PC samples were used for elemental analysis. The NanoMoudi II with aluminum foil as a collection substrate can be used to collect particles with a reasonable size distribution, and the aluminum foil and PC filters (on cellulose filters) have comparable collection efficiencies (Fujitani et al., 2006). Therefore, we assumed that the NanoMoudi II-Au and NanoMoudi II-PC collected particles with reasonable size distributions, and their size distributions were comparable with each other. In this paper, the data for the backup filters are not presented because their collection characteristics (especially adsorption of gaseous compounds) are remarkably different from those of the impaction substrates.

## 2.5 Particle mass

The particle masses of NanoMoudi II-Au and NanoMoudi II-PC samples were determined from the differences between the weights of the collection substrates before and after the sampling. For the NanoMoudi II-PC samples, only PC filters were weighed after eliminating static electricity using an ion balancer (TAS-182 NWM, Trinc Corp., Shizuoka, Japan). The substrates were weighed with a microbalance (readability 0.1 μg, UMX 2, Mettler-Toledo, Columbus, OH, USA) in a chamber (CHAM-1000, Horiba) in which the temperature and relative humidity were controlled at 21.5 °C and 35%, respectively. Each sample was weighed twice, and the obtained results were averaged. If the difference between two

recorded weights exceeded 0.5 µg in the case of the Au samples or 2.0 µg in the case of the PC samples, the sample was re-weighed. The samples were not conditioned before the weighing because Au foils and PC filters have low hygroscopicity.

## 2.6 EC/OC

The EC, OC, and total carbon (TC) in the NanoMoudi II-Au samples were determined by using a thermal/optical carbon analyzer (DRI Model 2001 Carbon Analyzer; Desert Research Institute, Las Vegas, NV, USA) (Chow et al., 1993). Three-eighths of each Au-foil sample cut in a fan shape was analyzed after the outside of the deposit area (diameter: 28 mm) had been cut off. The samples were analyzed using the IMPROVE protocol ($OC_1$: 120 °C; $OC_2$: 250 °C; $OC_3$: 450 °C; $OC_4$: 550 °C [in a 100% He atmosphere]; $EC_1$: 550 °C; $EC_2$: 700 °C; $EC_3$: 800 °C [in a 2% $O_2$/98% He atmosphere] (Chow et al., 2001)). The pyrolysis of the OC during analysis was not corrected because adequate correction using laser light is not possible with Au-foil samples.

## 2.7 Elements

The elemental compositions of the NanoMoudi II-PC samples were determined using particle-induced X-ray emission (PIXE) analysis at the Nishina Memorial Cyclotron Center of the Japan Radioisotope Association in Iwate, Japan. The target elements were Na, Mg, Al, Si, P, S, Cl, K, Ca, Mn, Fe, Co, Ni, Cu, Zn, Ga, As, Se, Br, Sr, Y, Zr, Nb, Mo, Hg, and Pb. The NanoMoudi II-PC samples were mounted on a Mylar target frame and bombarded with 2.9-MeV protons from a small cyclotron (Sera et al., 1992). The beam current, accumulated charge, and typical measuring time were 40–60 nA, 20–58 µC, and 10–12 min, respectively. The X-ray spectra thus obtained were analyzed using the SAPIX program (Sera et al., 1992). A quantitative analysis of the elemental values was performed using the Nuclepore-Br method (Sera et al., 1997). Blank filters were analyzed in all the procedures. The accuracy of the PIXE analysis was confirmed based on the National Institute of Standards and Technology (NIST) standard reference materials (Saitoh and Sera, 2005).

## 2.8 Organic composition

The organic compounds in the NanoMoudi II-Au samples were analyzed using thermal desorption gas chromatography/mass spectrometry (TD-GC/MS), which is sensitive and suitable for trace-level particulate samples (Fushimi et al., 2011). A thermal desorption unit (TDU; Gerstel GmbH & Co. KG, Mülheim an der Ruhr, Germany), a cooled injection system as a GC inlet (CIS 4; Gerstel), 6890 GC (Agilent Technologies, Palo Alto, CA, USA), and a double-focusing magnetic sector mass spectrometer (JMS-700K, JEOL, Tokyo, Japan) were used. For the GC columns, a DB-5MS (length: 30 m, internal diameter: 0.25 mm, film thickness: 0.25 µm; Agilent Technologies, Palo Alto, CA, USA) was used.

The NanoMoudi II-Au samples were cut into a fan shape (1/8–3/8 of 28 mm Φ, PM mass per sample: 1–20 µg) and were placed in a glass liner (length: 60 mm, outside diameter: 6 mm, inside diameter: 5 mm, Gerstel). The samples were transferred into the TDU and 1 µL of the internal standard mixture of $^{13}$C-labeled polycyclic aromatic hydrocarbons (approximately 0.5 µg mL$^{-1}$ for each compound; US EPA 16 PAH cocktail, Cambridge Isotope Laboratories (CIL), Andover, MA, USA) and deuterium $n$-alkane mixtures (10.7 µg mL$^{-1}$ of $C_{24}D_{50}$ $n$-alkane and 11.0 µg mL$^{-1}$ of $C_{30}D_{62}$ $n$-alkane) were injected onto the surface of the samples using an autosampler (MPS-TEX, Gerstel). The samples were then thermally desorbed using the TDU; the temperature was increased from 30 °C (held for 0.5 min) to 350 °C (held for 3 min) at 50 °C min$^{-1}$, using a helium desorption flow at 50 mL min$^{-1}$ in splitless mode. The interface temperature was maintained at 350 °C. During desorption at the TDU, the desorbed compounds were focused at 0 °C on a quartz wool inside the glass liner (inside diameter: 2 mm) in the CIS 4 for subsequent GC/MS analysis. The CIS 4 was then programmed to increase the temperature from 0 °C (held for 0.75 min) to 150 °C at 960 °C min$^{-1}$ and from 150 °C to 350 °C (held for 3 min) at 720 °C min$^{-1}$ to inject focused compounds into the GC column. The injection was performed in splitless mode with a 3-min splitless time. The GC oven was programmed to increase the temperature from 40 °C (held for 3 min) to 150 °C at 20 °C min$^{-1}$ and to

320 °C at 10 °C min$^{-1}$ (held for 15 min). Helium was used as a carrier gas at 2.5 mL min$^{-1}$ in a constant flow mode. The temperature of the transfer line between the GC and MS was 320 °C. The samples were ionized using the electron ionization method (ionizing voltage: 70 V, ionizing current: 600 μA, ion source temperature: 260 °C). The MS was operated in scan mode (*m/z* 35–400) with a mass resolution of 1000 to obtain comprehensive information regarding the organic compounds in

the particulate samples. The accelerating voltage was 10.0 kV, and the detector voltage was 0.40 kV.

The Jet A1 fuels and the jet lubrication oils were diluted by approximately 1000 times with *n*-hexane (dioxin analytical grade, Wako Pure Chemical Industries) and then analyzed with TD-GC/MS under the same condition as the particulate samples.

## 3 Results and discussion

### 3.1 Particle number size distribution

In our parallel measurements at the measurement site, the size distribution and concentrations measured using the EEPS agreed well with those measured using the SMPS for particles larger than 15 nm. A typical example is shown in Fig. S1. However, the EEPS can show an artifact peak at approximately 10 nm with polydisperse particles, which is not usually observed in the case of the SMPS (Fujitani et al., 2012). Therefore, we treat the EEPS data below 10 nm as supporting information and indicate it using dashed lines in this paper.

From the measurements obtained using the EEPS, total particle number concentrations remarkably increased up to more than $1 \times 10^6$ particles cm$^{-3}$ consistently when an aircraft took off or landed while wind (northerly or easterly winds) was blowing from the runway to the measurement site (Fig. 2). The peak concentrations of total particle number ($6.8 \times 10^5 - 1.3 \times 10^6$ particles cm$^{-3}$) during the plume event (indicated in Fig. 2) are approximately two orders of magnitude higher than the baseline concentrations ($1.1–1.6 \times 10^4$ particles cm$^{-3}$) during the no-plume event. Most peaks of particle number

concentrations seem to be attributed to specific take-offs or landings of aircraft, judging from the synchronized increase in $CO_2$ (data are not presented in this paper) and the reasonable time delay (approximately 20–200 s) between aircraft take-off/landing and the increase in particle number concentration. A one-month time series of total number concentrations and size distribution of particles is shown in Fig. S2. The total particle number concentrations during operation hours (6:00–23:00) were remarkably higher than those during non-operational hours (23:00–6:00). There was no noticeable enhancement

of nanoparticles during non-operational hours.

The total number concentrations at our measurement site are higher than the maximum value ($1.5 \times 10^5$ particles cm$^{-3}$) measured within 3 km of Los Angeles International (LAX) Airport (Hudda et al., 2014). The result seems reasonable because our measurement site is much closer to the runway (i.e., 140 m). In fact, Zhu et al. (2011) reported higher total particle number concentrations (i.e., $>1 \times 10^7$ particles cm$^{-3}$) during take-offs at the blast fence of the LAX airport. The total

particle number concentrations at our measurement site is higher than the average concentrations at a roadside with a large amount of heavy-duty diesel vehicles in Kawasaki, Japan in winter 2011 ($1.2 \times 10^5$ particles cm$^{-3}$ (Fujitani et al., 2012)).

Typical examples of the size distributions of particle number concentrations measured with an engine exhaust particle sizer (EEPS) are shown in Fig. 3A. When the aircraft exhaust plume approached the measurement site, the modal diameters were approximately 10 nm or smaller; these values are smaller than those of diesel vehicle exhaust particles (Fushimi et al., 2011).

In contrast, without the aircraft exhaust plumes, nucleation-mode particles were not observed. The number size distributions at 40–500 nm with aircraft exhaust plumes did not show a significant difference as compared to those without aircraft exhaust plumes. The mass size distributions, as estimated from the measured number concentrations while assuming a density of 1.0 g cm$^{-3}$, showed significant enhancements in the nucleation mode and a slight increase in the accumulation mode (Fig. 3B). These results clearly indicate that nanoparticles of diameters <30 nm contribute to the major fraction of

aircraft exhaust particle mass.

The size distributions of particle number concentrations averaged during the sampling periods are shown in Fig. S3 (A). During the daytime sampling periods, the modal diameters were approximately 10 nm or smaller. In contrast, during the nighttime sampling period, the modal diameter was 34 nm, and the peak concentrations observed ($1.1 \times 10^4$ particles cm$^{-3}$) were lower than those observed during the daytime ($8.8$–$21 \times 10^4$ particles cm$^{-3}$) by one order of magnitude. These results clearly show that aircraft emissions greatly affect the atmosphere at our sampling site during the daytime sampling periods. Although the air was not necessarily transported from the runway to the measurement site for the entire daytime sampling periods, remarkable enhancement of nanoparticles was observed on multiple days during every daytime sampling period. We estimated the particle mass concentrations from the measured number concentrations while assuming a density of 1.0 g cm$^{-3}$ (Fig. S3 (B, C)). At the nanoparticle size range (stage 11–13), the particle mass concentrations during the daytime were several times those observed during the nighttime. This suggests that aircraft emissions also greatly affect the nanoparticle samples (stage 11–13) during the daytime on mass basis.

## 3.2 Particle mass and chemical components

The mass concentrations of the particles, OC, EC, sulfur and other elements are shown in Fig. 4 by particle size. Generally, particle mass concentrations showed bimodal distributions with fine mode (diameter: 0.11–0.56 μm) and coarse mode (1.0–9.9 μm). The concentrations of the total particle mass were 9.7–13.4 μg m$^{-3}$ during the daytime and 10.7 μg m$^{-3}$ during the nighttime. At the nanoparticle stage (S11–S13, diameter: 10–57 nm), 4–10 μg of the particulate samples per stage were collected, which was assumed to be sufficient for the chemical analyses. The size distributions of particle mass concentrations of NanoMoudi II-AU and NanoMoudi II-PC agreed reasonably with each other.

The OC showed not only a fine mode at 0.11–1.0 μm but also a nucleation mode at 18–57 nm for the daytime samples #1 and #2. The sulfur also showed a bimodal distribution with a nucleation mode at 10–32 nm and a fine mode in all the daytime samples. In contrast, the EC showed a monomodal distribution with a modal diameter of 0.11–0.56 μm.

In this study, inorganic salts such as nitrate and ammonium were not measured. Thus, large percentages of the particulate mass remained unidentified, especially in the fine and coarse modes. On the other hand, in the nanoparticle size range, NanoMoudi II may have positive artifact on the particulate mass because more than half of the particulate mass often remains unidentified with the measured OC, EC, ions, and elements even in the case of vehicle exhaust particles, of which the major components are supposed to be carbonaceous (Fushimi et al., 2011; Fushimi et al., 2016). In addition, the particle mass concentration at each stage in the nanoparticle size range was approximately 1.4–10 times those estimated from the number concentrations measured using the EEPS (Figs. S3C and 4). Therefore, we assumed that the general chemical characteristics of the nanoparticles can be explained based on the measured components.

The proportions of OC, EC, sulfur, and other elements are shown in Fig. 5 by particle size. For the daytime samples #1 and #2, the EC percentages were high (up to >40%) at approximately 0.057–0.29 μm (stage 8–10). In contrast, the smaller particles in the nanoparticle size range comprised larger percentages of OC. The OC proportions were approximately 60–80% in the nanoparticles (10–32 nm) of the daytime samples #1 and #2. The sulfur and the sum of the other elements comprised approximately <10% and 10–30%, respectively, in the nanoparticles (10–32 nm) of the daytime samples #1 and #2. These results suggest that the nanoparticles emitted from a wide variety of in-use commercial aircraft with high thrusts mainly consist of OC. This is interesting and important because there has been some inconsistency between the volatility of the nanoparticles and the composition of the bulk particles in previous studies. However, if aircraft exhaust nanoparticles primarily comprise OC, the higher volatility appears to be reasonable. The OC proportions can be larger (EC proportions can be smaller) if the pyrolysis that occurs during the carbon analysis is corrected. Furthermore, the masses of the organic materials are generally 1.2–3.1 times that of the OC (Bae et al., 2006), and sulfur and other elements often exist as organic or

inorganic compounds. Therefore, the proportion of organic materials, sulfur compounds, and other organic/inorganic elements would be larger than that presented in this paper.

**3.3 Organic composition of size-resolved particles**

A chemical analysis of the size-resolved ambient particles sampled during daytime suggests that nanoparticles having diameters of 10–32 nm (S12 and S13 of the impactor stages) mainly comprise organic carbon (see supplement for details). We thus focus on the chemical characteristics of organic compounds for this size range, in comparison with those of jet fuels and jet lubrication oils.

The mass chromatograms (*m/z* 85) of size-resolved particles collected during the daytime (7:00–22:00, aircraft operation hours) and nighttime (0:00–6:00, non-operation hours), jet lubrication oil, and Jet A1 fuel are shown in Fig. 6. A series of *n*-alkanes was detected in the mass chromatograms (*m/z* 85, an indicator of hydrocarbons) of Jet A1 fuels (Fig. 6). The carbon numbers of these *n*-alkanes were in the range of $C_{11}$–$C_{18}$, with the largest peak for $C_{14}$. Interestingly, "humps" (baseline elevations) in the mass chromatograms at *m/z* 85—which is often detected in mineral-oil-based lubricants commonly used for automobiles—were not detected in jet lubrication oils (Mobil Jet Oil II and Mobil Jet Oil 254). Instead, approximately 25 distinct peaks (likely fatty acid esters of pentaerythritol) were detected at the retention time of approximately 21–29 min (corresponding to molecular weights of approximately 380–530) from two jet lubrication oils (Fig. 6). This is considered to be reasonable because the base stocks of jet lubrication oils are essentially a mixture of $C_5$–$C_{10}$ fatty acid esters of pentaerythritol (Timko et al., 2010; Yu et al., 2012). Furthermore, three compound groups (N-phenyl-1-naphthylamine, alkylated diphenyl amines, and tricresyl phosphate) were detected, which are reported as toxic substances in the material safety data sheet of lubrication oils. These three compound groups and fatty acid esters can be used as good markers for jet lubrication oil or jet exhaust because they are not usually contained in mineral-oil-based lubricants. In fact, these markers for jet lubrication oil were not detected from a mineral-oil-based lubricant for diesel vehicles with our TD-GC/MS analysis. *n*-Alkanes were not detected from two jet lubrication oils.

The daytime nanoparticle samples (S13: 10–18 nm, S12: 18–32 nm, and S11: 32–57 nm) clearly show the presence of oil-marker peaks at a retention time of approximately 21–27 min (likely fatty acid esters of pentaerythritol). The intensity ratios of these peaks at 22.3 min or later were very similar to those of a jet lubrication oil (Fig. 7). However, the intensities of the peaks at 22.3 min or earlier in the case of the daytime samples were lower than those of a jet lubrication oil. This may be due to the partial evaporation of more volatile compounds in the atmosphere, which was found to be the case for diesel exhaust nanoparticles in roadside atmospheres (Fushimi et al., 2008; Harrison et al., 2016). The mass spectra of peaks at 21–27 min in nanoparticles collected during the daytime were very similar to that of a jet lubrication oil (an example is shown in Fig. 8). In contrast, the oil-marker peaks were very small in larger particles collected during the daytime (e.g., S9: 105–170 nm or S7: 290–560 nm) and were not detected in nanoparticles collected during the nighttime (Fig. 6). The other marker compounds for jet lubrication oil mentioned above were also detected in the daytime nanoparticle samples but not in the nighttime samples. $C_{22}$–$C_{33}$ *n*-alkanes were detected in the daytime S9 and S7 samples but were weak or not detected in nanoparticles.

**4 Conclusions and implications**

From the aforementioned results, we conclude that approximately half the organic compounds in the <30-nm particles detectable using TD-GC/MS can be attributed to nearly intact forms of jet lubrication oil. This has not been identified in previous studies. Jet lubrication oil is released into the atmosphere through a centrifugal breather vent, located in bypass air flow, as a droplet smaller than approximately 1 μm or as vapor (Timko et al., 2010). The vented lubrication oil may be mixed with hot combustion gas at the exhaust area or in the atmosphere.

Our findings have an important implication for environmental issues from the ground level to the upper troposphere. The development of superior technologies for controlling oil emissions (e.g., through a breather vent) may greatly reduce aircraft exhaust nanoparticles. A reduction in the oil contributions would be beneficial in mitigating the health risk caused by aircraft exhaust nanoparticles as jet lubrication oils contain some toxic materials. Furthermore, a detailed knowledge of aircraft emissions is also required for improving our understanding of the origin and fate of ambient particles in the upper troposphere, which can potentially affect the radiative balance of the atmosphere (Righi et al., 2016).

We believe ambient measurements, such as those described in this paper, can provide complementary insights into aircraft emissions that are not obtained from engine exhaust measurements. However, our ambient particulate samples may have been affected by emissions from a wide variety of jet engines, operating conditions (e.g., take-off, landing, taxing, and idling), maintenance conditions, and other sources (e.g., auxiliary power units) (Stettler et al., 2011; Yu et al., 2012; Masiol and Harrison, 2014). The chemical composition of nanoparticles is also the average of a variety of emissions, although take-off and landing seems to have a great impact because the measurement site is near the runway. These emissions should be separately evaluated in the future.

**Data availability**

For the data shown in this paper, please contact the corresponding author via email (fushimi.akihiro@nies.go.jp).

**Supplement**

The supplement related to this article is available online at: https://doi.org/.........

**Author contributions**

AF contributed to conceptualization, validation, investigation, resources, data curation, original manuscript draft preparation, manuscript review and editing, visualization, and supervision. KS contributed to conceptualization, validation, investigation, resources, data curation, and manuscript review and editing. YF contributed to validation, investigation, resources, data curation, and manuscript review and editing. NT contributed to conceptualization, validation, investigation, data curation, manuscript review and editing, supervision, project administration, and funding acquisition.

**Competing interests**

The authors declare no competing interests.

**Acknowledgements**

This work was supported by the Environment Research and Technology Development Fund (5-1709) of the Ministry of the Environment, Japan. We thank Narita International Airport Corporation and Narita International Airport Promotion Foundation for helping with the field measurements at the Narita International Airport. We also thank Mr. Takumi Saotome of the Research Institute for Environmental Strategies, Inc. for providing the required jet fuel and lubrication oil; Dr. Hiromu Sakurai and Ms. Yoshiko Murashima of the National Institute of Advanced Industrial Science and Technology for evaluating the performance of the particle number measurements; Prof. Koichiro Sera of Iwate Medical University for his assistance with the PIXE analysis; Dr. Shunji Hashimoto, Dr. Teruyo Ieda, Dr. Kiyoshi Tanabe, and Dr. Yoshikatsu Takazawa of the

National Institute for Environmental Studies (NIES) for their valuable comments on our TD-GC/MS analysis; Ms. Maki Chiba and Ms. Masayo Ihara of NIES for assisting in the filter preparation, weighing, and carbon analysis; Dr. Kentaro Misawa of Tokyo Metropolitan University, Mr. Masato Nakamura of Green Blue Corp., and Mr. Hidenori Konno of Horiba Techno Service Ltd. for their assistances with the sampling; Mr. Yutaka Sugaya of NIES for assistance with the EEPS data analysis; and Mr. Ikuo Terabayashi of Aomori prefecture quantum science center for his valuable comments on the jet engine mechanism.

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

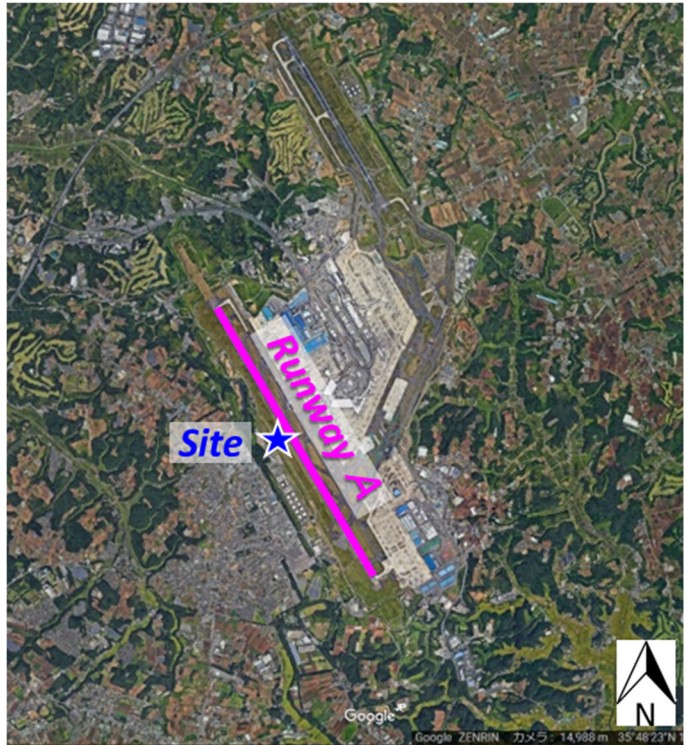

**Figure 1.** Map of the measurement site.

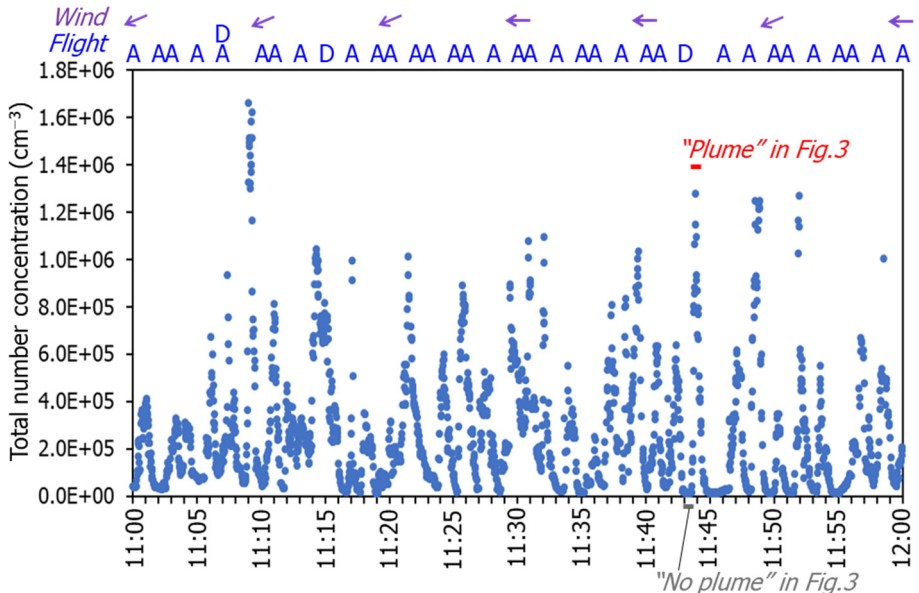

**Figure 2.** Time series total number concentrations of particles measured using the EEPS between 11:00–12:00 on February 15, 2018. One-second data is shown. The letters "D" and "A" indicate the departure and arrival times of aircraft fleets, respectively, reported by Narita International Airport. The time periods of the "Plume" and the "No plume" events, shown in Fig. 3, are shown here as bars. Wind directions measured at the measurement site are shown as arrows. The wind speed was 3.2–4.8 m s$^{-1}$ during this hour.

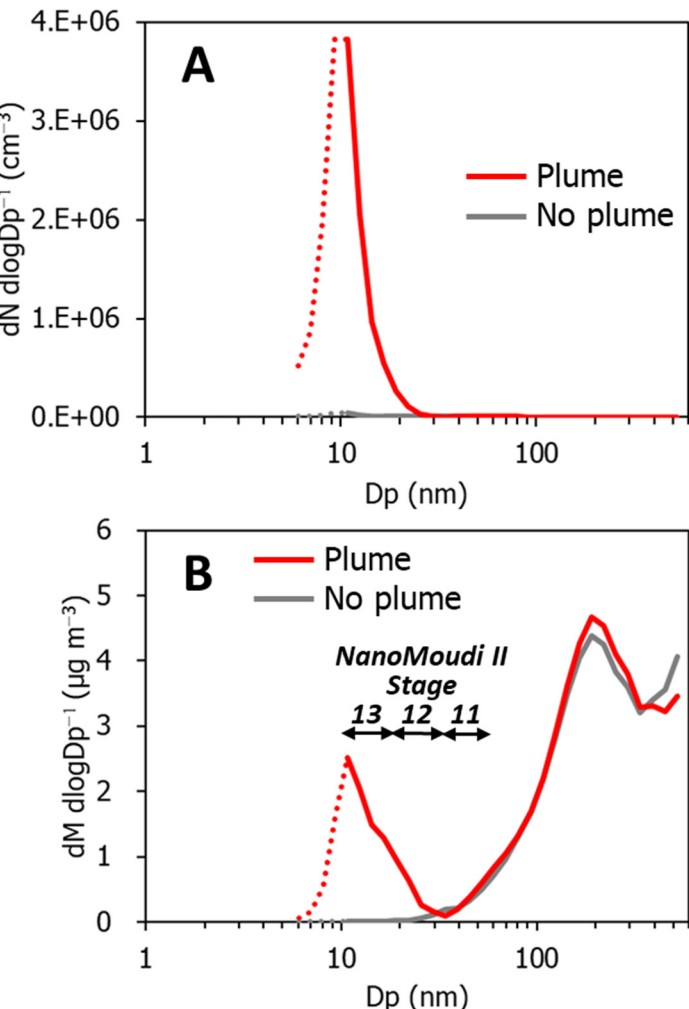

**Figure 3.** Size distributions of particle concentrations with and without aircraft exhaust plumes. (A) Particle number concentrations. (B) Estimated particle mass concentrations. The averaging times are approximately 20 s for both events. The measurement periods were 11:43:42–11:44:01 on February 15, 2018 and 11:43:11–11:43:30 on February 15, 2018 for the "plume" and "no plume" events, respectively, as indicated in Fig. 2.

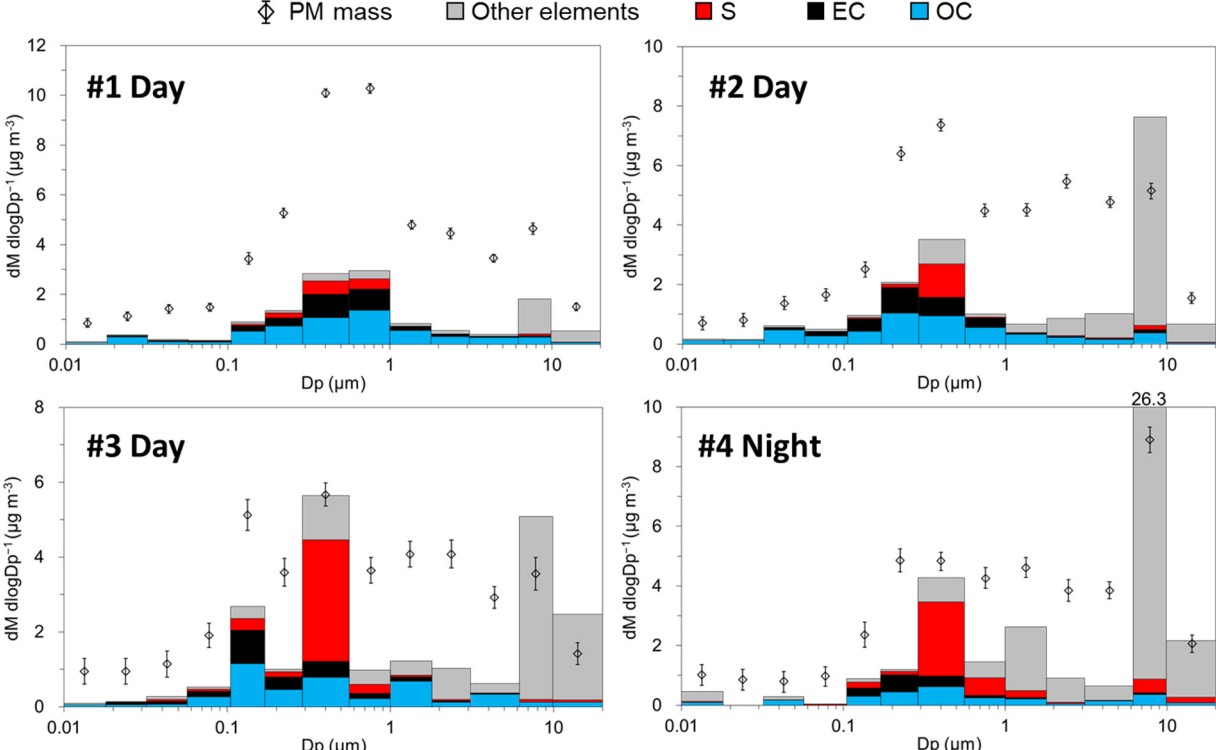

**Figure 4.** Mass concentrations of particles, OC, EC, sulfur, and other elements by particle size. The data of sulfur and other elements were adjusted so that the particulate masses of the NanoMoudi II-PC samples at each stage are equal to that of the NanoMoudi II-Au samples.

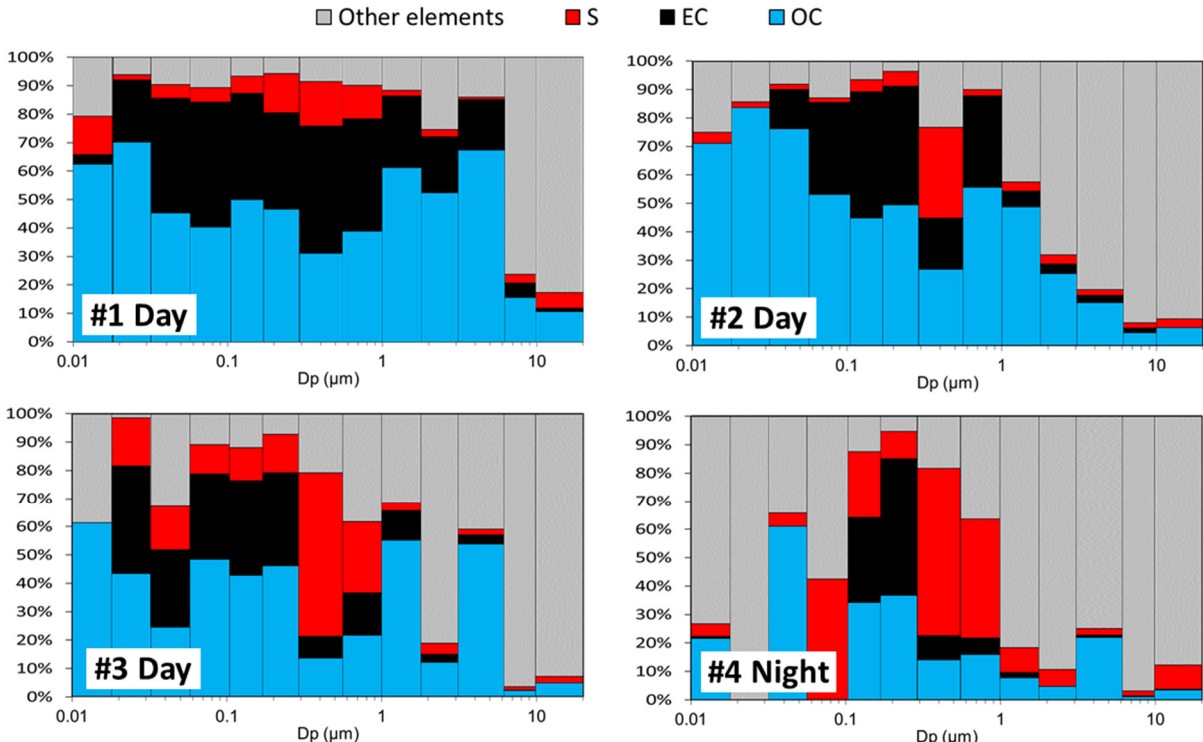

**Figure 5.** Proportions of OC, EC, sulfur, and other elements by particle size.

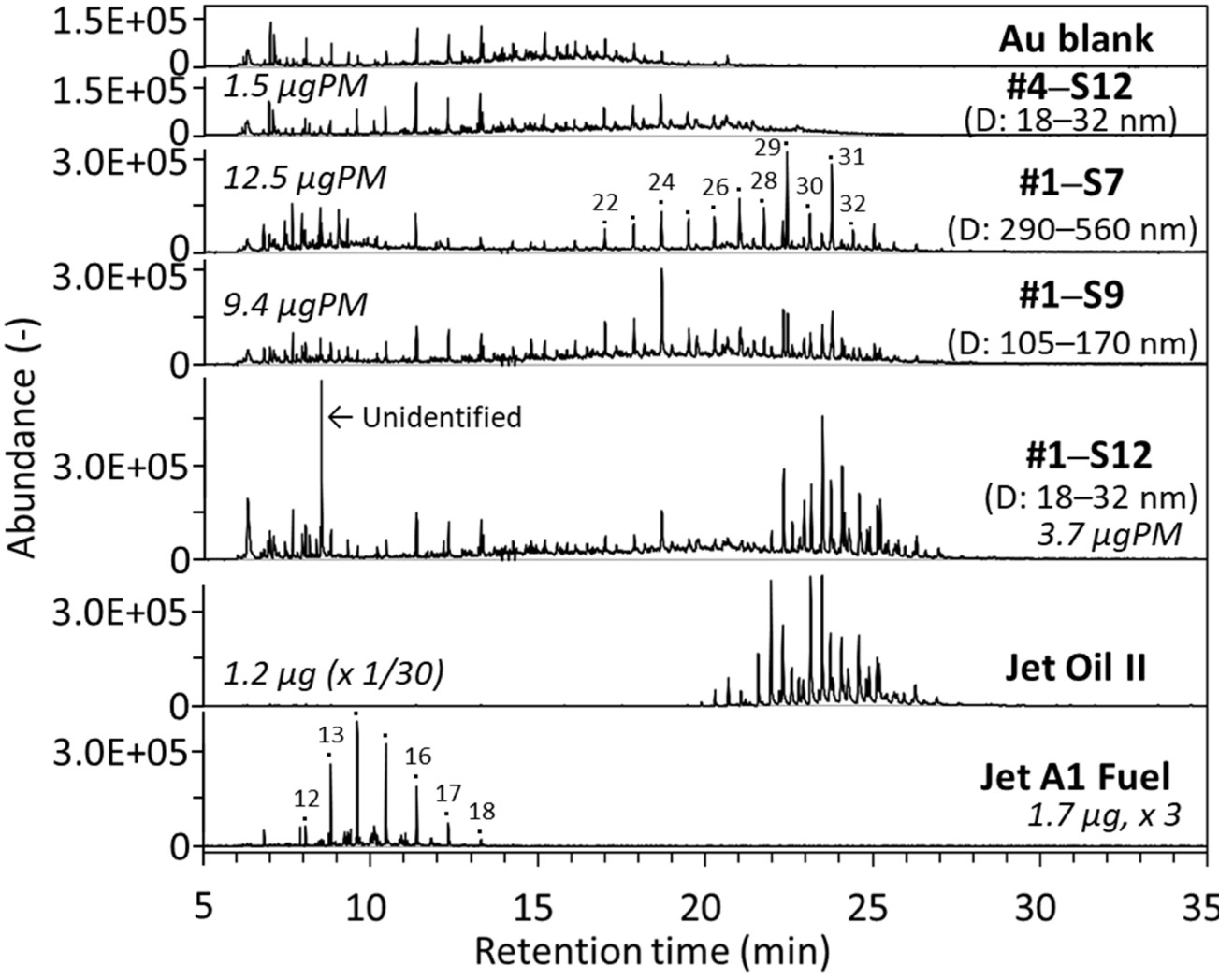

**Figure 6.** Mass chromatograms (m/z 85) of size-resolved ambient particles collected during the daytime (sample #1, February 9–13, 2018) and nighttime (sample #4, February 22–26, 2018) at the Narita International Airport. Mass chromatograms of Jet Oil II, Jet A1 fuel, and Au-foil blank are also shown for comparison. The mass values (in μg) presented in the plots indicate the mass of the samples that were analyzed using TD-GC/MS. The carbon numbers of *n*-alkanes are shown in the chromatograms of the S7 particles and Jet A1 fuel.

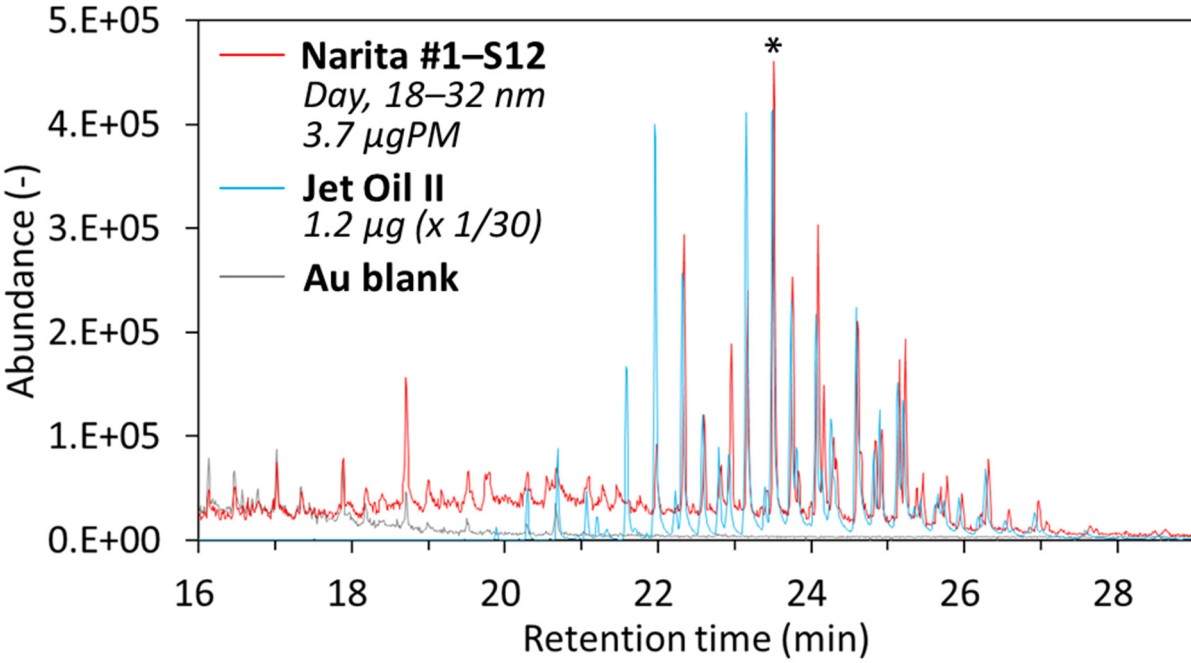

**Figure 7.** Mass chromatograms (m/z 85) of a nanoparticle sample collected during the daytime (sample #1, February 9–13, 2018; S12, diameter: 18–32 nm) at the Narita International Airport, Jet Oil II, and Au-foil blank. The mass spectra of the peaks at 23.51 min of Narita #1–S12 sample and Jet Oil II with an asterisk are shown in Fig. 8.

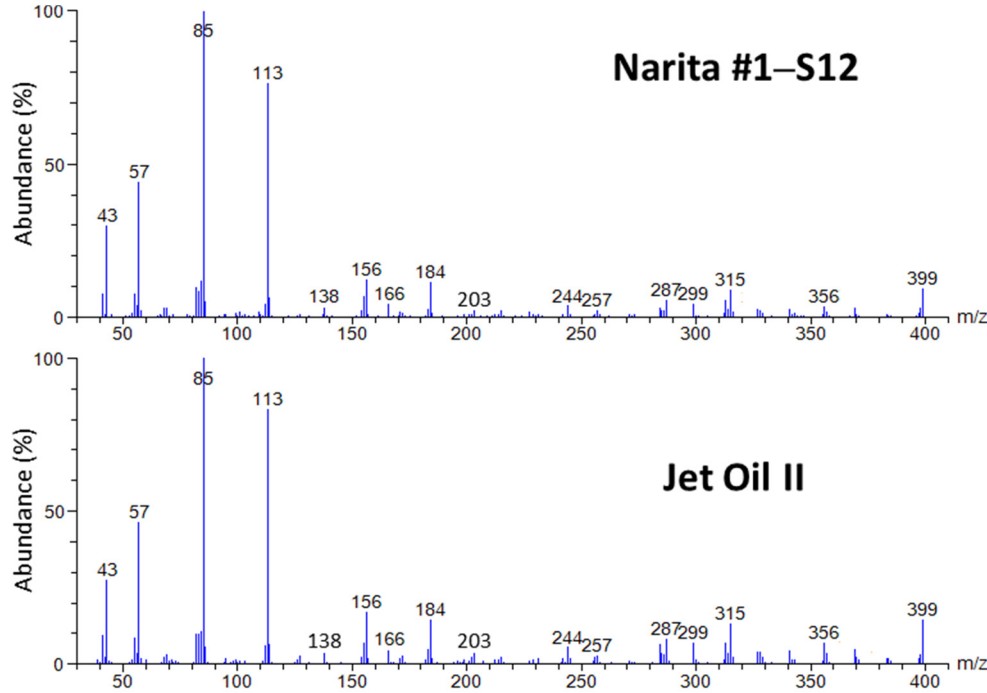

**Figure 8.** Mass spectra of a nanoparticle sample collected during the daytime at the Narita International Airport (sample #1, February 9–13, 2018; S12, diameter: 18–32 nm) and Jet Oil II at 23.51 min of the TD-GC/MS chromatograms. The chromatographic peaks are indicated in Fig. 7. The instrument background spectrum was subtracted for the Narita #1–S12 sample.

