# Peer review of "Identification of jet lubrication oil as major component of aircraft exhaust nanoparticles"

_Atmospheric Chemistry and Physics, 2018_

## Short Comment (SC1) · 14 Feb 2019

1) Designs of the venting of lubrication oil systems for different aircraft engines can vary dramatically. Some engines vent oil system excess air to the bottom of the nacelle into ambient air, while some other engines vent directly into the engine combustion exhaust at high temperature. Thus, according to the previous studies, the contribution from lubrication oil to aircraft organic PM emissions for different engines could vary from 5% to almost 100%. Providing a detailed description of the aircraft fleet as well as the associated engine types could be very helpful for the readers to understand and evaluate the obtained measurement results. 2) Engine operational conditions such as engine power could have significant influences on lubrication oil emissions. Impacts of PM emissions from aircraft on local air quality are normally evaluated during the

landing and take-off (LTO) cycles. Given the locations of sampling at the airport and wind directions, an estimate of the contribution of each stage during the LTO cycle (taxi, takeoff, and landing may affect different sampling locations) to the oil emissions would be helpful. 3) Oil emissions could also be associated with engine maintenance. One of the challenges of evaluating aviation PM emissions is the lack of information on engine maintenance, which results in a large discrepancy in emission measurements even on the same type of the aircraft engines. Could the authors comment on this issue?

---

## Referee Comment (RC1) · Anonymous Referee #1 · 25 Feb 2019

Fushimi et al. present a study on the chemical analysis of nanoparticles in aircraft engine exhaust. They identify jet lubrication oil as a major component of particles < 30 nm. The size resolved sampling using a multi-stage cascade impactor took place near the Narita International airport. Chemical analysis of the samples used a thermal desorption unit coupled to gas chromatography /mass spectrometry. The authors designed the study with necessary caution, in terms of duplicate sampling, dilution of the sampling flow for measurements of particle number concentration, analyzing nighttime (airport non-flight hours) vs daytime samples and analyzing reference jet fuel and lubrication oils. The usually challenging task of chemical analysis of nanoparticles (limited mass) is overcome in this case by high particle number concentrations.

The measured particle number concentration brings me to my main point of concern

regarding this paper: although, one might expect high particle number concentration near an airport, so far I have not seen ambient number concentrations of 2.5e7 cm-3 (Figure S3). The authors should comment on the reason of these high numbers, and should point out the differences to other studies: e.g. Hudda et al. observed max. 1.5e5 cm-3 within 3 km of LAX airport. Also, it would be helpful to show a longer time series of particle number measurements additionally to Figure S3 (which is only one hour). This might shed light into certain conditions that favor the buildup of such extremely high nanoparticle number concentrations. For such a time series graph a log y-axis might be helpful in order to evaluate the urban background particle number concentrations – if these numbers are reasonable, the measured numbers at Narita Intl. airport during operation hours would appear more plausible.

Apart from this point, the study appears well designed and the chemical analysis using GC/MS is solid. The main conclusion – identification of lubrication oil - is robust, since comparison by retention time, m/Z and EI-fragmentation spectra agree well with pure jet lubrication oil products.

Minor/technical comments:

p.1 l.28: Please comment on the formation of soot under idling vs. low- and high-thrust.

p.3 l.7-10: Did you calculate/estimate the inlet line losses of the small particles?

p.3. l.25: Did the manufacturer recommend the backup filters used (two different ones)? Can the differences of the back-up filters alter flow rates and thus cut-off sizes?

p. 3 l. 28: A nitrocellulose filter installed underneath the PC filter? Does this have an effect on the distance between nozzle and impaction plate?

p.4 l.1: These numbers are slightly different from the specs of the manufacturer. Did you measure cut-off sizes? If yes, how?

p.4 l.19 The experimental section should describe the procedure of the gravimetric analysis in the main text, since this is extremely challenging for small size fractions.

[Figure]

p.5 l.19: OC is roughly 2/3, and only during day. There is also 15-20% sulfur in one sample, and another nighttime sample is 100% "other elements". Hence, the statement that nanoparticles comprise "mainly organic carbon" is not correct.
* * *

---

## Referee Comment (RC2) · Anonymous Referee #2 · 6 Mar 2019

The paper by Fushimi et al. presents an investigation of the chemical composition of ultrafine particles measured close to a runway of the Narita International Airport, Japan. Several studies have already shown that large amounts of particles smaller than 30 nm are emitted by aircrafts. However, hardly anything is known about their chemical composition. Since these particles have only little mass, it is very difficult to investigate their composition. The chemical composition, however, is important to know for estimating how relevant aircraft emissions are for public health and whether they have an impact on weather or climate. Knowing where exactly these particles originate would also give the opportunity to find solutions for reducing their emissions. The authors find that the organic compounds of the particles measured near the runway are dominated by nearly intact forms of jet engine lubrication oil. Knowing this can help

to develop techniques for controlling oil emissions, which could greatly reduce aircraft exhaust particles.

The subject of this paper is relevant for ACP. Overall, the experimental data are presented well and sound. I recommend publication of this paper after the authors address my comments.

General comment:

It is known that aircraft emissions largely differ for changing jet engine conditions (e.g. Masiol and Harrison, 2014). The largest amount of particles may not necessarily be emitted while take-off (high thrust). Instead, particle emissions can be also very high while taxing or just while running the APU. From the map it looks like the measurement site could not only receive the particle emissions from take-off or landing. Since the taxi track and the terminal area are just behind the runway, the measured particles could also well be dominated by other jet engine conditions. The authors need to discuss how much their results on particle composition may depend on these different engine conditions. To my point of view, this is important with regard to the conclusion that is drawn – I think one cannot be sure that the lubricant oil particles are also emitted to a reasonable amount in the upper troposphere, so it is questionable whether they "can potentially affect the radiative balance of the atmosphere".

Specific comments:

1) What was the wind direction and wind speed during each measurement? Was the air transported from the runway to the measurement site for the entire measurement period?

2) I would strongly recommend shifting the appendix into the main text because it is important information.

3) Fig. 1: How do the authors define a "plume"? Can the peaks in number concentration be attributed to specific take-offs or landing aircrafts? What is the fraction of

"plume" to "no-plume" events?

Technical comments:

1) p 1, l 14, "A new particulate...": This sentence is without any context.

2) p 6, l 4, "The mass chromatograms...": This sentence makes more sense to shift a few lines above, e.g., to p 5, l 21.

3) Fig. 1: It does not say here what the dotted line means.

4) Fig. 2: I find the scale of the y-axis somehow weird. Should it not range from zero to 2.5e6 for every single subplot?

References:

Masiol, M. and Harrison, R. M. (2014): Aircraft engine exhaust emissions and other airport-related contributions to ambient air pollution: A review, Atm. Environ., 95, 409-455, https://doi.org/10.1016/j.atmosenv.2014.05.070.

---

## Author Comment (AC1) · 19 Apr 2019

**Authors' comments (final response) to the referees' comments and the other short comments**

Submitted to: Atmos. Chem. Phys.

Ms. Ref. No.:    acp-2018-1351

Title: Identification of jet lubrication oil as major component of aircraft exhaust nanoparticles

Authors: Akihiro Fushimi, et al.

Thank you very much for your constructive and valuable comments on our manuscript. We have fully taken them into account in revising our manuscript as detailed below (Our response is shown **in blue**). The revised contents in the revised manuscript are also shown **in blue**.

**RC1: Anonymous Referee #1**

*RC1: 'Review: Identification of jet lubrication oil as major component of aircraft exhaust nanoparticles',
Anonymous Referee #1, 25 Feb 2019*

**General comments**

*(Referee #1 comment)* Fushimi et al. present a study on the chemical analysis of nanoparticles in aircraft engine exhaust. They identify jet lubrication oil as a major component of particles < 30 nm. The size resolved sampling using a multi-stage cascade impactor took place near the Narita International airport. Chemical analysis of the samples used a thermal desorption unit coupled to gas chromatography /mass spectrometry. The authors designed the study with necessary caution, in terms of duplicate sampling, dilution of the sampling flow for measurements of particle number concentration, analyzing nighttime (airport non-flight hours) vs daytime samples and analyzing reference jet fuel and lubrication oils. The usually challenging task of chemical analysis of nanoparticles (limited mass) is overcome in this case by high particle number concentrations.

*(Author's response)* Thank you very much for your positive comments.

*(Referee #1 comment)* The measured particle number concentration brings me to my main point of concern regarding this paper: although, one might expect high particle number concentration near an airport, so far I have not seen ambient number concentrations of 2.5e7 cm-3 (Figure S3). The authors should comment on the reason of these high numbers, and should point out the differences to other studies: e.g. Hudda et al. observed max. 1.5e5 cm-3 within 3 km of LAX airport.

*(Author's response)* (1) We have rechecked the data (total number concentration shown in original Fig. S3) and found those data was incorrect.

(Author's changes in manuscript) We have corrected the data and replaced the figure (Fig. 2). Now, total number concentration became 1/16 of the original values, and the maximum concentration is about 1.7E+06 cm$^{-3}$.

We added the following explanation. "The total number concentrations at our measurement site is higher than the maximum value ($1.5 \times 10^5$ particles cm$^{-3}$) measured within 3 km of Los Angeles International (LAX) Airport (Hudda et al., 2014). It seems reasonable because our measurement site is much closer to the runway (i.e. 140 m). In fact, Zhu et al. (2011) reported higher total particle number

concentrations (i.e. >1 × 10$^7$ particles cm$^{-3}$) during takeoffs at the blast fence of LAX airport." (P6, L28-31)

Accordingly, we added the following reference.

- Zhu, Y., Fanning, E., Yu, R. C., Zhang, Q., Froines, J. R.: Aircraft emissions and local air quality impacts from takeoff activities at a large International Airport, Atmos. Environ., 45, 6526-6533 (2011).

(Referee #1 comment) Also, it would be helpful to show a longer time series of particle number measurements additionally to Figure S3 (which is only one hour). This might shed light into certain conditions that favor the buildup of such extremely high nanoparticle number concentrations. For such a time series graph a log y-axis might be helpful in order to evaluate the urban background particle number concentrations – if these numbers are reasonable, the measured numbers at Narita Intl. airport during operation hours would appear more plausible.

(Author's response) We agree with the reviewer.

(Author's changes in manuscript) We added a figure of the one-month time series total number concentrations and size distribution of particles. (Fig. S2) Accordingly, we added related explanation about the figure. "One-month time series total number concentrations and size distribution of particles is shown in Fig. S2. Total particle number concentrations during operation hours (6:00–23:00) were remarkably larger than non-operational hours (23:00–6:00). Also, there was no noticeable enhancement of nanoparticles during non-operational hours." (P6, L24-27)

(Referee #1 comment) Apart from this point, the study appears well designed and the chemical analysis using GC/MS is solid. The main conclusion – identification of lubrication oil - is robust, since comparison by retention time, m/z and EI-fragmentation spectra agree well with pure jet lubrication oil products.

(Author's response) Thank you very much for your positive comments.

**Minor/technical comments:**

(Referee #1 comment) p.1 l.28: Please comment on the formation of soot under idling vs. low- and high-thrust.

(Author's response) We agree.

(Author's changes in manuscript) We added some explanation about low thrust, idling, and high thrust conditions shown below. "low engine thrust (e.g. idle and taxi modes)" and "high engine thrust (e.g. take-off and climb)" (P2, L1).

(Referee #1 comment) p.3 l.7-10: Did you calculate/estimate the inlet line losses of the small particles?

(Author's response) Yes.

(Author's changes in manuscript) We added the following sentence. "For the SMPS, the penetration efficiency through the sampling tube was estimated to be >70% approximately above 10 nm based on the theoretical formulae of Gormley and Kennedy (1949)." (P3, L17-19).

(Referee #1 comment) p.3. l.25: Did the manufacturer recommend the backup filters used (two different ones)? Can the differences of the back-up filters alter flow rates and thus cut-off sizes?

(Author's response) The default filter substrates for the back-up of the samplers (NanoMoudiI, MSP) are glass fiber filter and quartz fiber filter.

(Author's changes in manuscript) We added the following explanation. "Using different substrates of the backup filters do not alter the flow rates." (P4, L6)

(Referee #1 comment) p. 3 l. 28: A nitrocellulose filter installed underneath the PC filter? Does this have an effect on the distance between nozzle and impaction plate?
(Author's response) A nitrocellulose filter was installed underneath the PC filter.
(Author's changes in manuscript) We replaced the word "under" to "underneath" (P4, L3). When a nitrocellulose filter was installed underneath the PC filter, the distance between nozzle and impaction plate change slightly. However, the effect to the change in cut-off size seems negligible because of the results we got. To explain this, we added following description in the manuscript. "on cellulose filters" (P4, L16)

(Referee #1 comment) p.4 l.1: These numbers are slightly different from the specs of the manufacturer. Did you measure cut-off sizes? If yes, how?
(Author's response) We did not measure the cut-off sizes. The cut-off sizes of each sampler (NanoMoudiII, MSP) are calibrated and reported with each sampler by the manufacture.
(Author's changes in manuscript) We added the following explanation. ", calibrated and reported by the manufacture," (P4, L8).

(Referee #1 comment) p.4 l.19 The experimental section should describe the procedure of the gravimetric analysis in the main text, since this is extremely challenging for small size fractions.
(Author's response) We agree.
(Author's changes in manuscript) According to the comments from you and the Referee #2, we moved most information originally written in the supplement, including the method to obtain particle mass (original Supplement, Section 1.1), to the main body of the manuscript.

(Referee #1 comment) p.5 l.19: OC is roughly 2/3, and only during day. There is also 15-20% sulfur in one sample, and another nighttime sample is 100% "other elements". Hence, the statement that nanoparticles comprise "mainly organic carbon" is not correct.
(Author's response) The original sentence was not correct. The explanation was intended to be only for "Daytime" samples not for "Nighttime samples".
(Author's changes in manuscript) We added "daytime" in the sentence (P8, L28). For the daytime samples, organic carbon is the primary components. So, we did not change the other explanation.

*RC2: 'Review of Fushimi et al.', Anonymous Referee #2, 06 Mar 2019*

**General comments**

(Referee #1 comment) The paper by Fushimi et al. presents an investigation of the chemical composition of ultrafine particles measured close to a runway of the Narita International Airport, Japan. Several studies have already shown that large amounts of particles smaller than 30 nm are emitted by aircrafts. However, hardly anything is known about their chemical composition. Since these particles have only little mass, it is very difficult to investigate their composition. The chemical composition, however, is important to know for estimating how relevant aircraft emissions are for public health and whether they have an impact on weather or climate. Knowing where exactly these particles originate would also give the opportunity to find solutions for reducing their emissions. The authors find that the organic compounds of the particles measured near the runway are dominated by nearly intact forms of jet engine lubrication oil. Knowing this can help to develop techniques for controlling oil emissions, which could greatly reduce aircraft exhaust particles.

The subject of this paper is relevant for ACP. Overall, the experimental data are presented well and sound. I recommend publication of this paper after the authors address my comments.

(Author's response) Thank you very much for your positive comments.

(Referee #2 comment) It is known that aircraft emissions largely differ for changing jet engine conditions (e.g. Masiol and Harrison, 2014). The largest amount of particles may not necessarily be emitted while take-off (high thrust). Instead, particle emissions can be also very high while taxing or just while running the APU. From the map it looks like the measurement site could not only receive the particle emissions from take-off or landing. Since the taxi track and the terminal area are just behind the runway, the measured particles could also well be dominated by other jet engine conditions. The authors need to discuss how much their results on particle composition may depend on these different engine conditions.

(Author's response) This is a very important point and should be evaluated in the future.

(Author's changes in manuscript) We added the following explanation at the "Conclusions and implications" chapter. "We believe ambient measurements, like shown in this paper, can provide complementary insights into aircraft emissions which are not obtained from engine exhaust measurements. However, our ambient particulate samples may have affected by emissions from wide variety of jet engines, operating conditions (e.g. take-off, landing, taxiing and idling), maintenance conditions, and other sources (e.g. auxiliary power units and ground service equipments) (Stettler et al., 2011; Yu et al., 2012; Masiol and Harrison, 2014). The chemical composition of nanoparticles is also the average of those variety of emissions, although take-off and landing seems to have greater impact because the measurement site is near the runway. Those emissions should be separately evaluated in the future." (P10, L10-16).

(Referee #2 comment) To my point of view, this is important with regard to the conclusion that is drawn – I think one cannot be sure that the lubricant oil particles are also emitted to a reasonable amount in the upper troposphere, so it is questionable whether they "can potentially affect the radiative balance of the atmosphere".

(Author's response) As suggested by the referee, it is bit uncertain that the lubrication-oil-dominated nanoparticles are actually emitted from aircraft in the upper troposphere. However, the sentence in the original manuscript was intended to describe possible importance of aircraft PM emission in the upper troposphere, which is apart from its chemical composition.

(Author's changes in manuscript) Therefore, we did not change the words here. (P10, L9)

**Specific comments**

(Referee #2 comment) 1) What was the wind direction and wind speed during each measurement? Was the air transported from the runway to the measurement site for the entire measurement period?

(Author's changes in manuscript) We added the wind roses and the data of wind speed during each sampling period in Fig. S2. We added the following sentence "Although the air was not necessarily transported from the runway to the measurement site for the entire sampling period for the daytime samples, remarkable enhancement of nanoparticles was observed multiple days in every daytime sampling period." (P7, L21-23).

(Referee #2 comment) 2) I would strongly recommend shifting the appendix into the main text because it is important information.

(Author's response) We agree.

(Author's changes in manuscript) We moved all the text and references in the original supplement to the main body of the manuscript (Sections 2.5, 2.6, 2.7, 3.1, and 3.2). Original Figs. S2 and S4 (now Figs. S1 and S3), and newly added figure and table (Fig. S2 and Table S1) were kept in the Supplement.

(Referee #2 comment) 3) Fig. 1: How do the authors define a "plume"? Can the peaks in number concentration be attributed to specific take-offs or landing aircrafts? What is the fraction of "plume" to "no-plume" events?

(Author's response) Regarding original Fig.1 (now Fig.3), we chose the "plume event" during an increasing event of number concentrations (total number concentrations were in the range of $6.8 \times 10^5 - 1.3 \times 10^6$ particles cm$^{-3}$) which has typical particle size distribution of plume events. We chose the "no-plume event" which has low and stable (like a baseline) total number concentrations ($1.1-1.6 \times 10^4$ particles cm$^{-3}$). Both events shown here have same duration (19 seconds). If we take the original Fig. S3 (Fig. 2 in the revised manuscript. 11:00-12:00 on February 15) as an example, the ratio of the time duration of plume event and non-plume event is about 3:1.

(Author's changes in manuscript) We added the following explanation "The peak concentrations of total particle number ($6.8 \times 10^5 - 1.3 \times 10^6$ particles cm$^{-3}$) during the plume event (indicated in Fig. 2) are about two orders of magnitude higher than the baseline concentrations ($1.1-1.6 \times 10^4$ particles cm$^{-3}$) during the no-plume event." in the manuscript (P6, L19-22).

Accordingly, in Fig.2, we indicated the periods for "plume event" and "no-plume event" of Fig.3.

We also added the wind vectors and information of take-off/landing in Fig.2. We added the following explanation "Judging from synchronized increase of $CO_2$ (data is not shown in this paper) and the reasonable time-delay (20–200 seconds approximately) between aircraft take-off/landing and increase of particle number concentration, it seems that most peaks of particle number concentrations can be attributed to specific take-offs or landings of aircraft fleets." in the manuscript (P6, L22-24).

Please also see our response to your general comments written above.

**Technical comments**

(Referee #2 comment) 1) p 1, l 14, "A new particulate: : :": This sentence is without any context..

(Author's response) We agree.

(Author's changes in manuscript) We added the following sentence. "Therefore, research needs to characterize aircraft exhaust particles have been increasing." (P1, L17-18).

(Referee #2 comment) 2) p 6, l 4, "The mass chromatograms: : :": This sentence makes more sense to shift a few lines above, e.g., to p 5, l 21..

(Author's response) We agree.

(Author's changes in manuscript) The sentence was moved to the place where the referee suggested. (P9, L1-2).

(Referee #2 comment) 3) Fig. 1: It does not say here what the dotted line means..

(Author's response) It was bit hard to find the explanation about this since it was originally written in the supplement.

(Author's changes in manuscript) We have moved most information in the original supplement to the main body of the manuscript. Accordingly, the following explanation about the referee's suggestion was also moved to the main text. "However, the EEPS can show an artifact peak at approximately 10 nm with polydisperse particles, which is not usually observed in the case of the SMPS (Fujitani et al., 2012). Therefore, we treat the EEPS data below 10 nm as supporting information and indicate it using dashed lines in this paper." (P6, L14-16).

(Referee #2 comment) 4) Fig. 2: I find the scale of the y-axis somehow weird. Should it not range from zero to 2.5e6 for every single subplot?.

(Author's response) We agree.

(Author's changes in manuscript) In the figure, we added the baseline at zero signal. We also added y-scale for every single subplot. (Fig.6).

**Reference**

(from the Referee #2) Masiol, M. and Harrison, R. M. (2014): Aircraft engine exhaust emissions and other airport-related contributions to ambient air pollution: A review, Atm. Environ., 95, 409-455, https://doi.org/10.1016/j.atmosenv.2014.05.070

(SC1-1) Designs of the venting of lubrication oil systems for different aircraft engines can vary dramatically. Some engines vent oil system excess air to the bottom of the nacelle into ambient air, while some other engines vent directly into the engine combustion exhaust at high temperature. Thus, according to the previous studies, the contribution from lubrication oil to aircraft organic PM emissions for different engines could vary from 5% to almost 100%. Providing a detailed description of the aircraft fleet as well as the associated engine types could be very helpful for the readers to understand and evaluate the obtained measurement results.

(Author's response) We agree.

(Author's changes in manuscript) We added a table which show the summary of aircraft models used in the Runway A of Narita International Airport, Japan during the measurement period (Table S1). Accordingly, the following explanation was added. "Summary of aircraft models used in the Runway A of Narita International Airport during our measurement period is shown in Table S1." (P2, L27-29)

(SC1-2) Engine operational conditions such as engine power could have significant influences on lubrication oil emissions. Impacts of PM emissions from aircraft on local air quality are normally evaluated during the landing and take-off (LTO) cycles. Given the locations of sampling at the airport and wind directions, an estimate of the contribution of each stage during the LTO cycle (taxi, takeoff, and landing may affect different sampling locations) to the oil emissions would be helpful.

(Author's response) This is a very important point and should be evaluated in the future.

(Author's changes in manuscript) We added the following explanation at the "Conclusions and implications" chapter. "We believe ambient measurements, like shown in this paper, can provide complementary insights into aircraft emissions which are not obtained from engine exhaust measurements. However, our ambient particulate samples may have affected by emissions from wide variety of jet engines, operating conditions (e.g. take-off, landing, taxiing and idling), maintenance conditions, and other sources (e.g. auxiliary power units and ground service equipments) (Stettler et al., 2011; Yu et al., 2012; Masiol and Harrison, 2014). The chemical composition of nanoparticles is also the average of those variety of emissions, although take-off and landing seems to have great impact because the measurement site is near the runway. Those emissions should be separately evaluated in the future." (P10, L10-16).

(SC1-3) Oil emissions could also be associated with engine maintenance. One of the challenges of evaluating aviation PM emissions is the lack of information on engine maintenance, which results in a large discrepancy in emission measurements even on the same type of the aircraft engines. Could the authors comment on this issue?

(Author's response) We do not have information about the engine maintenance so far.

(Author's changes in manuscript) We have added the explanation about the possible impact of maintenance condition in the manuscript as indicated in our response to your previous comment (SC1-2).

**Other revision**

(Author's response) We have added the following references according to the revision indicated above.

[revised manuscript text omitted]

**Figure S2.** One-month time series total particle number concentrations (PNC) and size distribution of particles measured using the EEPS in February 9–26, 2018. One-hour averaged data measured with EEPS is shown. The sampling periods and their wind roses at the measurement site are shown. The averages and the ranges (minimum – maximum) of the wind speeds during the sampling periods were 2.2 (0–9.2) m s$^{-1}$ (#1 Daytime), 2.9 (0–7.2) m s$^{-1}$ (#2 Daytime), 2.4 (0–6.9) m s$^{-1}$ (#3 Daytime), and 2.9 (0–6.1) m s$^{-1}$ (#4 Nighttime). The Runway A is in a northerly or easterly direction from our measurement site.

[Figure]

**Figure S3.** Size distributions of particle number concentrations measured using the EEPS averaged during the sampling periods (A). (B, C) Estimated size distributions of particle mass concentrations.

---

## Author Response (AR1)

**Authors' comments (*final response*) to the referees' comments and the other short comments (*April 24, 2019*)**

Submitted to: Atmos. Chem. Phys.

Ms. Ref. No.:    acp-2018-1351

Title: Identification of jet lubrication oil as major component of aircraft exhaust nanoparticles

Authors: Akihiro Fushimi, et al.

Thank you very much for your constructive and valuable comments on our manuscript. We have fully taken them into account in revising our manuscript as detailed below (Our response is shown **in blue**). The revised contents in the revised manuscript are also shown **in blue**.

*RC1: Anonymous Referee #1*

*RC1: 'Review: Identification of jet lubrication oil as major component of aircraft exhaust nanoparticles', Anonymous Referee #1, 25 Feb 2019*

**General comments**

*(Referee #1 comment)* Fushimi et al. present a study on the chemical analysis of nanoparticles in aircraft engine exhaust. They identify jet lubrication oil as a major component of particles < 30 nm. The size resolved sampling using a multi-stage cascade impactor took place near the Narita International airport. Chemical analysis of the samples used a thermal desorption unit coupled to gas chromatography /mass spectrometry. The authors designed the study with necessary caution, in terms of duplicate sampling, dilution of the sampling flow for measurements of particle number concentration, analyzing nighttime (airport non-flight hours) vs daytime samples and analyzing reference jet fuel and lubrication oils. The usually challenging task of chemical analysis of nanoparticles (limited mass) is overcome in this case by high particle number concentrations.

*(Author's response)* Thank you very much for your positive comments.

*(Referee #1 comment)* The measured particle number concentration brings me to my main point of concern regarding this paper: although, one might expect high particle number concentration near an airport, so far I have not seen ambient number concentrations of 2.5e7 cm-3 (Figure S3). The authors should comment on the reason of these high numbers, and should point out the differences to other studies: e.g. Hudda et al. observed max. 1.5e5 cm-3 within 3 km of LAX airport.

*(Author's response)* (1) We have rechecked the data (total number concentration shown in original Fig. S3) and found those data was incorrect.

(Author's changes in manuscript) We have corrected the data and replaced the figure (Fig. 2). Now, total number concentration became 1/16 of the original values, and the maximum concentration is about 1.7E+06 cm$^{-3}$.

We added the following explanation. "The total number concentrations at our measurement site are higher than the maximum value ($1.5 \times 10^5$ particles cm$^{-3}$) measured within 3 km of Los Angeles International (LAX) Airport (Hudda et al., 2014). The result seems reasonable because our measurement site is much closer to the runway (i.e., 140 m). In fact, Zhu et al. (2011) reported higher total particle number concentrations (i.e., $>1 \times 10^7$ particles cm$^{-3}$) during take-offs at

the blast fence of the LAX airport." (P5, L26-29)

Accordingly, we added the following reference.

-   Zhu, Y., Fanning, E., Yu, R. C., Zhang, Q., Froines, J. R.: Aircraft emissions and local air quality impacts from takeoff activities at a large International Airport, Atmos. Environ., 45, 6526-6533 (2011).

(Referee #1 comment) Also, it would be helpful to show a longer time series of particle number measurements additionally to Figure S3 (which is only one hour). This might shed light into certain conditions that favor the buildup of such extremely high nanoparticle number concentrations. For such a time series graph a log y-axis might be helpful in order to evaluate the urban background particle number concentrations – if these numbers are reasonable, the measured numbers at Narita Intl. airport during operation hours would appear more plausible.

(Author's response) We agree with the reviewer.

(Author's changes in manuscript) We added a figure of the one-month time series total number concentrations and size distribution of particles. (Fig. S2) Accordingly, we added related explanation about the figure. "A one-month time series of total number concentrations and size distribution of particles is shown in Fig. S2. The total particle number concentrations during operation hours (6:00–23:00) were remarkably higher than those during non-operational hours (23:00–6:00). There was no noticeable enhancement of nanoparticles during non-operational hours." (P5, L22-25)

(Referee #1 comment) Apart from this point, the study appears well designed and the chemical analysis using GC/MS is solid. The main conclusion – identification of lubrication oil - is robust, since comparison by retention time, m/z and EI-fragmentation spectra agree well with pure jet lubrication oil products.

(Author's response) Thank you very much for your positive comments.

**Minor/technical comments:**

(Referee #1 comment) p.1 l.28: Please comment on the formation of soot under idling vs. low- and high-thrust.

(Author's response) We agree.

(Author's changes in manuscript) We added some explanation about low thrust, idling, and high thrust conditions shown below. "low engine thrust (e.g. idle and taxi)" and "high engine thrust (e.g. take-off and climb)" (P1, L31).

(Referee #1 comment) p.3 l.7-10: Did you calculate/estimate the inlet line losses of the small particles?

(Author's response) Yes.

(Author's changes in manuscript) We added the following sentence. "For the SMPS, the penetration efficiency through the sampling tube was estimated to be >70% above 10 nm based on the theoretical formulae of Gormley and Kennedy (1949)." (P2, L38 - P3, L2).

(Referee #1 comment) p.3. l.25: Did the manufacturer recommend the backup filters used (two different ones)? Can the differences of the back-up filters alter flow rates and thus cut-off sizes?

(Author's response) The default filter substrates for the back-up of the samplers (NanoMoudiII, MSP) are glass fiber filter and quartz fiber filter.
(Author's changes in manuscript) We added the following explanation. "The use of different substrates as backup filters does not alter the flow rates." (P3, L21)

(Referee #1 comment) p. 3 l. 28: A nitrocellulose filter installed underneath the PC filter? Does this have an effect on the distance between nozzle and impaction plate?
(Author's response) A nitrocellulose filter was installed underneath the PC filter.
(Author's changes in manuscript) We replaced the word "under" to "underneath" (P3, L18). When a nitrocellulose filter was installed underneath the PC filter, the distance between nozzle and impaction plate change slightly. However, the effect to the change in cut-off size seems negligible because of the results we got. To explain this, we added following description in the manuscript. "(on cellulose filters)" (P3, L31)

(Referee #1 comment) p.4 l.1: These numbers are slightly different from the specs of the manufacturer. Did you measure cut-off sizes? If yes, how?
(Author's response) We did not measure the cut-off sizes. The cut-off sizes of each sampler (NanoMoudiII, MSP) are calibrated and reported with each sampler by the manufacture.
(Author's changes in manuscript) We added the following explanation. ", calibrated and reported by the manufacture," (P3, L23).

(Referee #1 comment) p.4 l.19 The experimental section should describe the procedure of the gravimetric analysis in the main text, since this is extremely challenging for small size fractions.
(Author's response) We agree.
(Author's changes in manuscript) According to the comments from you and the Referee #2, we moved most information originally written in the supplement, including the method to obtain particle mass (original Supplement, Section 1.1), to the main body of the manuscript.

(Referee #1 comment) p.5 l.19: OC is roughly 2/3, and only during day. There is also 15-20% sulfur in one sample, and another nighttime sample is 100% "other elements". Hence, the statement that nanoparticles comprise "mainly organic carbon" is not correct.
(Author's response) The original sentence was not correct. The explanation was intended to be only for "Daytime" samples not for "Nighttime samples".
(Author's changes in manuscript) We added "sampled during daytime" in the sentence (P7, L8). For the daytime samples, organic carbon is the primary components. So, we did not change the other explanation.

**General comments**

(Referee #1 comment) The paper by Fushimi et al. presents an investigation of the chemical composition of ultrafine particles measured close to a runway of the Narita International Airport, Japan. Several studies have already shown that large amounts of particles smaller than 30 nm are emitted by aircrafts. However, hardly anything is known about their chemical composition. Since these particles have only little mass, it is very difficult to investigate their composition. The chemical composition, however, is important to know for estimating how relevant aircraft emissions are for public health and whether they have an impact on weather or climate. Knowing where exactly these particles originate would also give the opportunity to find solutions for reducing their emissions. The authors find that the organic compounds of the particles measured near the runway are dominated by nearly intact forms of jet engine lubrication oil. Knowing this can help to develop techniques for controlling oil emissions, which could greatly reduce aircraft exhaust particles.

The subject of this paper is relevant for ACP. Overall, the experimental data are presented well and sound. I recommend publication of this paper after the authors address my comments.

(Author's response) Thank you very much for your positive comments.

(Referee #2 comment) It is known that aircraft emissions largely differ for changing jet engine conditions (e.g. Masiol and Harrison, 2014). The largest amount of particles may not necessarily be emitted while take-off (high thrust). Instead, particle emissions can be also very high while taxing or just while running the APU. From the map it looks like the measurement site could not only receive the particle emissions from take-off or landing. Since the taxi track and the terminal area are just behind the runway, the measured particles could also well be dominated by other jet engine conditions. The authors need to discuss how much their results on particle composition may depend on these different engine conditions.

(Author's response) This is a very important point and should be evaluated in the future.

(Author's changes in manuscript) We added the following explanation at the "Conclusions and implications" chapter. "We believe ambient measurements, such as those described in this paper, can provide complementary insights into aircraft emissions that are not obtained from engine exhaust measurements. However, our ambient particulate samples may have been affected by emissions from wide variety of jet engines, operating conditions (e.g., take-off, landing, taxing, and idling), maintenance conditions, and other sources (e.g., auxiliary power units) (Stettler et al., 2011; Yu et al., 2012; Masiol and Harrison, 2014). The chemical composition of nanoparticles is also the average of a variety of emissions, although take-off and landing seems to have a great impact because the measurement site is near the runway. These emissions should be separately evaluated in the future." (P8, L13-19).

(Referee #2 comment) To my point of view, this is important with regard to the conclusion that is drawn – I think one cannot be sure that the lubricant oil particles are also emitted to a reasonable amount in the upper troposphere, so it is questionable whether they "can potentially affect the radiative balance of the atmosphere".

(Author's response) As suggested by the referee, it is bit uncertain that the lubrication-oil-dominated nanoparticles are actually emitted from aircraft in the upper troposphere. However, the sentence in the original manuscript was intended to describe possible importance of aircraft PM emission in the upper troposphere, which is apart from its chemical composition.

(Author's changes in manuscript) Therefore, we did not change the words here. (P8, L12)

**Specific comments**

(Referee #2 comment) 1) What was the wind direction and wind speed during each measurement? Was the air transported from the runway to the measurement site for the entire measurement period?

(Author's changes in manuscript) We added the wind roses and the data of wind speed during each sampling period in Fig. S2. We added the following sentence "Although the air was not necessarily transported from the runway to the measurement site for the entire daytime sampling periods, remarkable enhancement of nanoparticles was observed during multiple days for each daytime sampling period." (P6, L11-12).

(Referee #2 comment) 2) I would strongly recommend shifting the appendix into the main text because it is important information.

(Author's response) We agree.

(Author's changes in manuscript) We moved all the text and references in the original supplement to the main body of the manuscript (Sections 2.5, 2.6, 2.7, 3.1, and 3.2). Original Figs. S2 and S4 (now Figs. S1 and S3), and newly added figure and table (Fig. S2 and Table S1) were kept in the Supplement.

(Referee #2 comment) 3) Fig. 1: How do the authors define a "plume"? Can the peaks in number concentration be attributed to specific take-offs or landing aircrafts? What is the fraction of "plume" to "no-plume" events?

(Author's response) Regarding original Fig.1 (now Fig.3), we chose the "plume event" during an increasing event of number concentrations (total number concentrations were in the range of $6.8 \times 10^5 - 1.3 \times 10^6$ particles cm$^{-3}$) which has typical particle size distribution of plume events. We chose the "no-plume event" which has low and stable (like a baseline) total number concentrations (1.1–1.6 × 10$^4$ particles cm$^{-3}$). Both events shown here have same duration (19 seconds). If we take the original Fig. S3 (Fig. 2 in the revised manuscript. 11:00-12:00 on February 15) as an example, the ratio of the time duration of plume event and non-plume event is about 3:1.

(Author's changes in manuscript) We added the following explanation "The peak concentrations of total particle number ($6.8 \times 10^5 - 1.3 \times 10^6$ particles cm$^{-3}$) during the plume event (indicated in Fig. 2) are approximately two orders of magnitude higher than the baseline concentrations (1.1–1.6 × 10$^4$ particles cm$^{-3}$) during the no-plume event." in the manuscript (P5, L17-20).

Accordingly, in Fig.2, we indicated the periods for "plume event" and "no-plume event" of Fig.3.

We also added the wind vectors and information of take-off/landing in Fig.2. We added the following explanation "Most peaks of particle number concentrations seem to be attributed to specific take-offs or landings of aircraft fleets, judging from the synchronized increase in CO$_2$ (data are

not presented in this paper) and the reasonable time delay (approximately 20–200 s) between aircraft take-off/landing and the increase in particle number concentration." in the manuscript (P5, L20-22).

Please also see our response to your general comments written above.

**Technical comments**

(Referee #2 comment) 1) p 1, l 14, "A new particulate: : :": This sentence is without any context..
(Author's response) We agree.
(Author's changes in manuscript) We added the following sentence. "Therefore, research needs to characterize aircraft exhaust particles have been increasing." (P1, L17-18).

(Referee #2 comment) 2) p 6, l 4, "The mass chromatograms: : :": This sentence makes more sense to shift a few lines above, e.g., to p 5, l 21..
(Author's response) We agree.
(Author's changes in manuscript) The sentence was moved to the place where the referee suggested. (P7, L12-13).

(Referee #2 comment) 3) Fig. 1: It does not say here what the dotted line means..
(Author's response) It was bit hard to find the explanation about this since it was originally written in the supplement.
(Author's changes in manuscript) We have moved most information in the original supplement to the main body of the manuscript. Accordingly, the following explanation about the referee's suggestion was also moved to the main text. "However, the EEPS can show an artifact peak at approximately 10 nm with polydisperse particles, which is not usually observed in the case of the SMPS (Fujitani et al., 2012). Therefore, we treat the EEPS data below 10 nm as supporting information and indicate it using dashed lines in this paper." (P5, L12-14).

(Referee #2 comment) 4) Fig. 2: I find the scale of the y-axis somehow weird. Should it not range from zero to 2.5e6 for every single subplot?.
(Author's response) We agree.
(Author's changes in manuscript) In the figure, we added the baseline at zero signal. We also added y-scale for every single subplot. (Fig.6).

**Reference**
(from the Referee #2) Masiol, M. and Harrison, R. M. (2014): Aircraft engine exhaust emissions and other airport-related contributions to ambient air pollution: A review, Atm. Environ., 95, 409-455, https://doi.org/10.1016/j.atmosenv.2014.05.070

(SC1-1) Designs of the venting of lubrication oil systems for different aircraft engines can vary dramatically. Some engines vent oil system excess air to the bottom of the nacelle into ambient air, while some other engines vent directly into the engine combustion exhaust at high temperature. Thus, according to the previous studies, the contribution from lubrication oil to aircraft organic PM emissions for different engines could vary from 5% to almost 100%. Providing a detailed description of the aircraft fleet as well as the associated engine types could be very helpful for the readers to understand and evaluate the obtained measurement results.

(Author's response) We agree.

(Author's changes in manuscript) We added a table which show the summary of aircraft models used in the Runway A of Narita International Airport, Japan during the measurement period (Table S1). Accordingly, the following explanation was added. "A summary of the aircraft models used in runway A of Narita International Airport during our measurement period is given in Table S1." (P2, L19-21)

(SC1-2) Engine operational conditions such as engine power could have significant influences on lubrication oil emissions. Impacts of PM emissions from aircraft on local air quality are normally evaluated during the landing and take-off (LTO) cycles. Given the locations of sampling at the airport and wind directions, an estimate of the contribution of each stage during the LTO cycle (taxi, takeoff, and landing may affect different sampling locations) to the oil emissions would be helpful.

(Author's response) This is a very important point and should be evaluated in the future.

(Author's changes in manuscript) We added the following explanation at the "Conclusions and implications" chapter. "We believe ambient measurements, such as those described in this paper, can provide complementary insights into aircraft emissions that are not obtained from engine exhaust measurements. However, our ambient particulate samples may have been affected by emissions from wide variety of jet engines, operating conditions (e.g., take-off, landing, taxing, and idling), maintenance conditions, and other sources (e.g., auxiliary power units) (Stettler et al., 2011; Yu et al., 2012; Masiol and Harrison, 2014). The chemical composition of nanoparticles is also the average of a variety of emissions, although take-off and landing seems to have a great impact because the measurement site is near the runway. These emissions should be separately evaluated in the future." (P8, L13-19).

(SC1-3) Oil emissions could also be associated with engine maintenance. One of the challenges of evaluating aviation PM emissions is the lack of information on engine maintenance, which results in a large discrepancy in emission measurements even on the same type of the aircraft engines. Could the authors comment on this issue?

(Author's response) We do not have information about the engine maintenance so far.

(Author's changes in manuscript) We have added the explanation about the possible impact of maintenance condition in the manuscript as indicated in our response to your previous comment (SC1-2).

**Other revision**

(Author's response) We have added the following references according to the revision indicated above.

[revised manuscript text omitted]

**Figure S2.** One-month time series of total particle number concentrations (PNCs) and size distribution of particles measured using the EEPS during February 9–26, 2018. One-hour averaged data are shown. The sampling periods and their wind roses at the measurement site are also shown. The averages and the ranges (minimum–maximum) of the wind speeds during the sampling periods were 2.2 (0–9.2) m s$^{-1}$ (#1 Daytime), 2.9 (0–7.2) m s$^{-1}$ (#2 Daytime), 2.4 (0–6.9) m s$^{-1}$ (#3 Daytime), and 2.9 (0–6.1) m s$^{-1}$ (#4 Nighttime). Runway A is in a northerly or easterly direction from our measurement site.

[Figure]

**Figure S3.** Size distributions of particle number concentrations measured using the EEPS averaged during the sampling periods (A). (B, C) Estimated size distributions of particle mass concentrations.

---

## Author Response (AR2)

**Author's Response to the Co-Editor Decision (Publish subject to technical corrections, 25 Apr 2019 by Dr. Joachim Curtius)**

**Akihiro Fushimi**

Submitted to: Atmos. Chem. Phys.

Ms. Ref. No.:   acp-2018-1351

Title: Identification of jet lubrication oil as major component of aircraft exhaust nanoparticles

Authors: Akihiro Fushimi, et al.

Thank you very much for your constructive and valuable comments on our manuscript. We have fully taken them into account in revising our manuscript as detailed below (Our response is shown **in blue**). The revised contents in the revised manuscript are also shown **in blue**.

*Co-Editor Decision: Publish subject to technical corrections (25 Apr 2019) by Dr. Joachim Curtius*

*(Co-Editor's comments)*

1.   pleases consider changing the title to:

"Identification of jet lubrication oil as *a* major component of aircraft exhaust nanoparticles"

2.   Also please include the following edits:

"particle number concentrations can be attributed to specific take-offs or landings of aircraft." (omit "fleets") (P6, L24).

3.   "...remarkable enhancement of nanoparticles was observed *on* multiple days *during*" every daytime sampling period." (P7, L23).

4.   "...our ambient particulate samples may have *been* affected by emissions from *a* wide variety of jet engines, operating conditions..."(P10, L12).

*(Author's response)* We agree to all the suggestions.

*(Author's changes in manuscript)* We have revised the manuscript according to the Co-Editor's suggestions as shown below.

1.   The title became to "Identification of jet lubrication oil as a major component of aircraft exhaust nanoparticles" (P1, L1 of the manuscript, and S1, L2 of the supplement).

[revised manuscript text omitted]

**Figure S2.** One-month time series of total particle number concentrations (PNCs) and size distribution of particles measured using the EEPS during February 9–26, 2018. One-hour averaged data are shown. The sampling periods and their wind roses at the measurement site are also shown. The averages and the ranges (minimum–maximum) of the wind speeds during the sampling periods were 2.2 (0–9.2) m s$^{-1}$ (#1 Daytime), 2.9 (0–7.2) m s$^{-1}$ (#2 Daytime), 2.4 (0–6.9) m s$^{-1}$ (#3 Daytime), and 2.9 (0–6.1) m s$^{-1}$ (#4 Nighttime). Runway A is in a northerly or easterly direction from our measurement site.

[Figure]

**Figure S3.** Size distributions of particle number concentrations measured using the EEPS averaged during the sampling periods (A). (B, C) Estimated size distributions of particle mass concentrations.